# Flexible supercapacitor electrodes based on real metal-like cellulose papers

Yongmin Ko[1], Minseong Kwon[1], Wan Ki Bae[2], Byeongyong Lee[3], Seung Woo Lee (ID) [3] & Jinhan Cho[1]

The effective implantation of conductive and charge storage materials into flexible frames has been strongly demanded for the development of flexible supercapacitors. Here, we introduce metallic cellulose paper-based supercapacitor electrodes with excellent energy storage performance by minimizing the contact resistance between neighboring metal and/or metal oxide nanoparticles using an assembly approach, called ligand-mediated layer-by-layer assembly. This approach can convert the insulating paper to the highly porous metallic paper with large surface areas that can function as current collectors and nanoparticle reservoirs for supercapacitor electrodes. Moreover, we demonstrate that the alternating structure design of the metal and pseudocapacitive nanoparticles on the metallic papers can remarkably increase the areal capacitance and rate capability with a notable decrease in the internal resistance. The maximum power and energy density of the metallic paper-based supercapacitors are estimated to be 15.1 mW cm$^{-2}$ and 267.3 µWh cm$^{-2}$, respectively, substantially outperforming the performance of conventional paper or textile-type supercapacitors.

[1] Department of Chemical and Biological Engineering, Korea University, 145 Anam-ro, Seongbuk-gu, Seoul 02841, Republic of Korea. [2] Photoelectronic Hybrids Research Center, Korea Institute of Science and Technology (KIST), Seoul 02792, Republic of Korea. [3] The George W. Woodruff School of Mechanical Engineering, Georgia Institute of Technology, Atlanta, Georgia 30332, USA. Correspondence and requests for materials should be addressed to S.W.L. (email: seung.lee@me.gatech.edu) or to J.C. (email: jinhan71@korea.ac.kr)

Flexible energy storage devices are a key enabling factor for the propagation of wearable or paper electronics in biomedical, consumer electronics, and military applications[1–5]. Lithium-ion batteries (LIBs) and supercapacitors are two commercial power sources that support consumer electronic devices. Supercapacitors play an increasing role in wearable devices due to their higher power density (>10 kW kg$^{-1}$), rapid charging/discharging, and longer operation lifetimes than LIBs[6–11]. However, the energy density of supercapacitors (~5 W h kg$^{-1}$) is significantly lower than that of LIBs (~150 W h kg$^{-1}$), and this parameter should be improved to meet the performance demand for next-generation flexible devices[6, 12]. However, fully flexible devices require the flexibility of energy storage devices in addition to conventional performance considerations, including high energy and high power and cycling stability. Additionally, areal performance parameters, such as areal capacitance (the capacitance per unit area, F cm$^{-2}$), along with power and energy densities, are also considered important performance indexes for flexible supercapacitors. Thus, the critical challenge in this field is to develop flexible supercapacitors with higher power and energy densities per unit area than those of existing energy storage devices.

One of the key components for flexible energy storage devices is a flexible and conductive substrate that can be used as a current collector. Papers and textiles have been considered ideal substrates due to their low cost, flexibility, and highly porous structures, which can absorb active electrode materials[1, 13]. However, these substrates must be coated with electrically conductive materials due to their insulating nature. Conductive materials, such as carbon nanotubes (CNTs) or silver nanowires, have been mainly incorporated into the paper or textile substrates to enhance their electric conductivity[1, 2]. Recently, a variety of pseudocapacitive (PC) materials with higher capacitances have been introduced to improve the energy density[14–17]. However, most of these PC nanoparticles (NPs) have low electrical conductivities, and thus the kinetic resistance of these NPs increases with the increasing loading amount (or layer thickness), significantly reducing the power density[18, 19]. Performance values close to the theoretical capacitance of PC NPs have been achieved by the formation of ultrathin PC NP arrays onto metal frames with high conductivity; however, these devices exhibited an extremely low areal capacitance due to the low mass loading of the active energy materials[20, 21].

Various physical adsorption processes (i.e., dip coating, painting, Meyer rod coating, and the dispensing-writing method) have been employed to incorporate the conductive and active materials into the paper or textile substrates[22–26]. However, an intrinsic limitation of the traditional physical adsorption processes is poor control over the loading amount of the conductive and active components due to the large surface roughness and porosity of the paper and textile substrates[1]. This limitation becomes critical when such materials are incorporated into microelectronics or microscale energy storage devices, which require precise control over the thickness or energy density. Thus, developing a versatile assembly process that can incorporate both conductive and active components into the flexible substrates with the precise control of loading (or capacity) is crucial for flexible supercapacitor applications. The layer-by-layer (LbL) assembly process is a simple fabrication technique used to precisely control the loading amount of the active materials in the various substrates based on the complementary interactions between the species, irrespective of

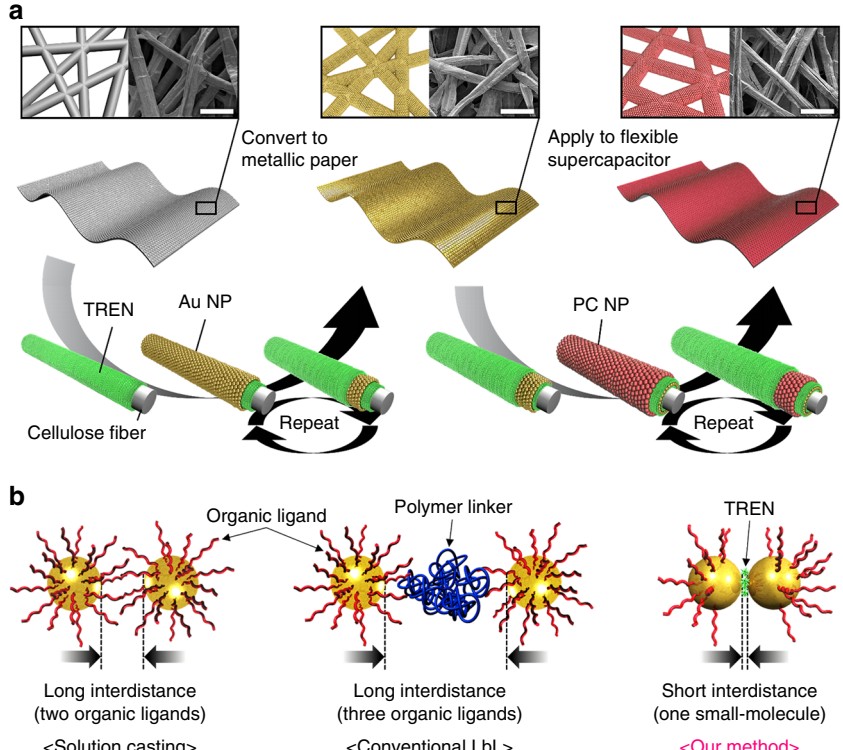

**Fig. 1** Metallic paper (MP)-based supercapacitor electrode. **a** Schematic for the preparation of the MP-based supercapacitor electrodes using ligand-mediated layer-by-layer (LbL) assembly between hydrophobic metal (or metal oxide) nanoparticles (NPs) and TREN molecules. In this case, the internal porous structure of the paper is perfectly preserved even after deposition of the NPs as shown in FE-SEM images. All *scale bars* in FE-SEM images indicate 50 μm. **b** Schematic diagram showing the difference between the conventional solution casting of the NPs (*left*), traditional LbL assembly (based on electrostatic interactions, hydrogen bonding, or covalent bonding) (*middle*), and our approach (i.e., molecule ligand-mediated LbL assembly) (*right*)

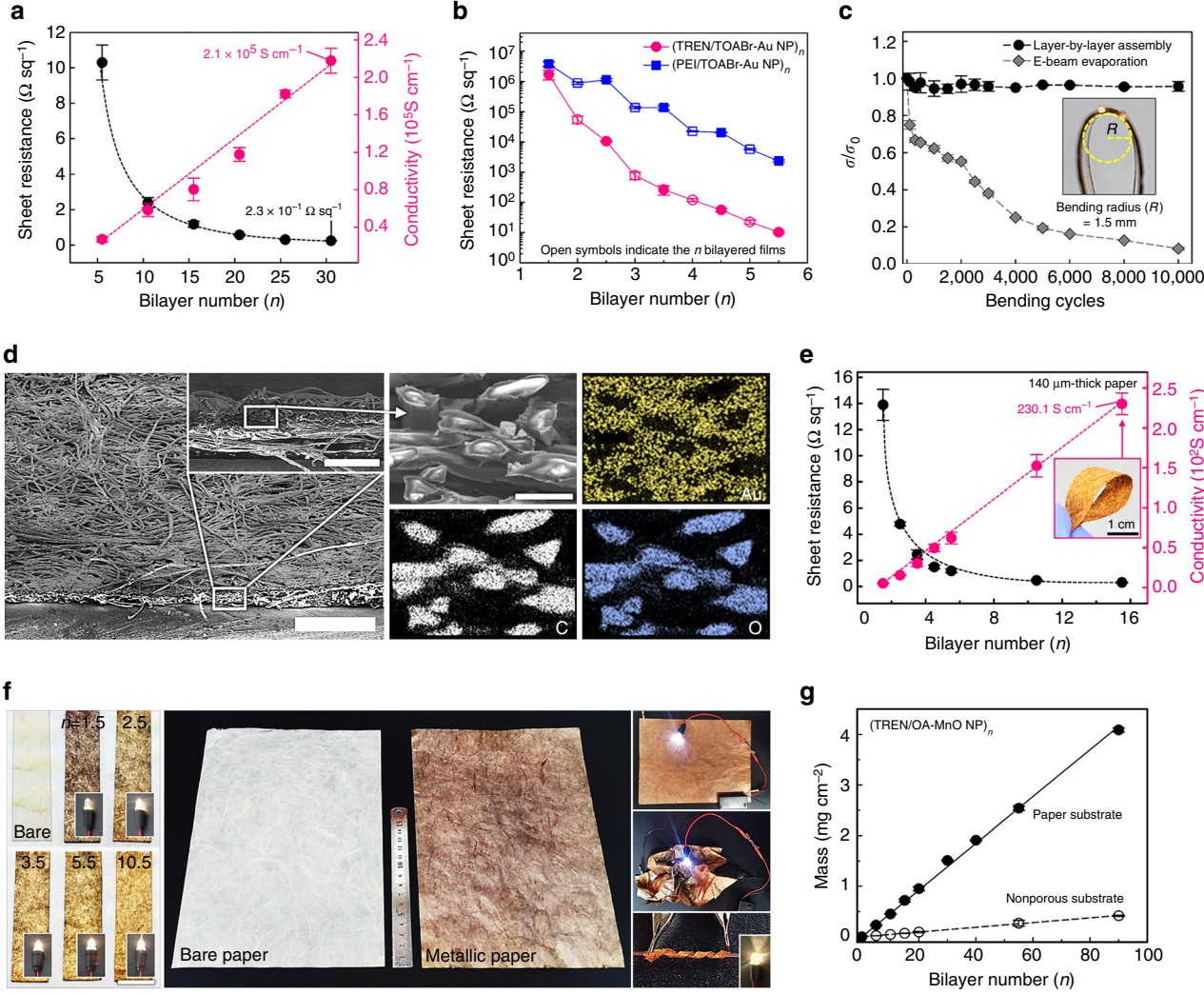

**Fig. 2** Physical properties of the (TREN/TOABr-Au NP)$_n$-coated electrode. **a** Sheet resistances and electric conductivities of the (TREN/TOABr-Au NP)$_n$ multilayers coated onto quartz glass as a function of the bilayer number ($n$). **b** Effect of the outermost organic ligands on the sheet resistances of the (TREN/TOABr-Au NP)$_n$ and (PEI/TOABR-Au NP)$_n$ multilayers coated onto quartz glass. **c** Mechanical stability tests for the (TREN/TOABr-Au NP)$_{15.5}$ multilayer- and the evaporated Au-coated PET substrates as a function of the bending cycling number (bending radius of ~1.5 mm). **d** Tilted and cross-sectional (*inset*) FE-SEM images of the (TREN/TOABr-Au NP)$_{10}$-coated paper (*scale bar*, 500 and 200 µm (*inset*)) (*left*) and EDX mapping images of the (TREN/TOABr-Au NP)$_{10}$-coated paper electrode (*white square* of the cross-sectional SEM image (*inset*)) (*scale bar*, 20 µm) (*right*). **e** Electrical properties of the (TREN/TOABr-Au NP)$_n$-coated paper with increasing bilayer number ($n$). **f** Photographic images of the (TREN/TOABr-Au NP)$_n$-coated paper with an LED connection as a function of the bilayer number ($n$) (*scale bar*, 1 cm) (*left*) and the large-area (20 cm × 30 cm) cellulose paper (*middle*). The images on the *right* indicate the LED connection between the MPs under various flexible conditions, such as flat, crumpling, and wrapping (using a glass stick with a diameter of 1.5 mm). **g** Mass loading of the (TREN/OA-MnO NP)$_n$ multilayers coated onto the porous paper electrodes with a thickness of 140 µm and the nonporous Au-coated QCM electrodes. The mass percentage of the MnO NPs within the multilayers measured from thermogravimetric analyses (TGA) was ~87%

the substrate size and shape[27–40]. Previous studies (mainly, electrostatic LbL assembly in aqueous solution) have successfully fabricated thin-film electrodes with controlled thickness and loaded a variety of electrochemically active materials, including functionalized CNTs, graphene oxides, conductive polymers, and metal oxides, for supercapacitor or LIB applications[31–33]. However, conventional electrostatic LbL assembly of NPs for energy storage electrodes has some limitations on (1) low packing density (<30%) of NPs in lateral dimension due to reciprocal electrostatic repulsion between same charged NPs, (2) high contact resistance among NPs, and (3) the blocking-up phenomena of voids within porous templates due to the use of bulky polymer linkers sandwiched between adjacent NP layers[41–43].

Herein, we introduce a high-performance and flexible metallic paper-based supercapacitor (MP-SC) that is fabricated by an assembly approach, i.e., ligand-mediated LbL assembly that can directly bridge all the interfaces of metal and/or metal oxide NPs through small molecules. For this study, the commercial cellulose papers are first changed into real metal-like papers (sheet resistance of MP: ~0.3 Ω sq$^{-1}$) through a molecule ligand-mediated LbL assembly of tetraoctylammonium bromide (TOABr)-stabilized Au NPs (TOABr-Au NPs), and tris(2-aminoethyl) amine (TREN). In this case, the bulky TOABr ligands are replaced by TREN molecules during LbL deposition. After such a ligand replacement, the formation of only one TREN molecule layer between the adjacent Au NP layers can significantly reduce the contact resistance between neighboring

NPs and, as a result, impart electrical properties similar to those of the bulk Au films. The electrical conductivity and sheet resistance of the pure (TREN/TOABr-Au NP)$_{30}$ multilayers without any additional treatments are measured to be $2.1 \times 10^5$ S cm$^{-1}$ and $0.23\ \Omega$ sq$^{-1}$, respectively, outperforming those of previously reported hybrid conductors[44–46]. The porous MPs are used as an effective reservoir for the incorporation of high-energy PC NPs (i.e., oleic acid-stabilized MnO (OA-MnO) or OA-Fe$_3$O$_4$ NPs) through the continuous ligand-mediated LbL assembly process. Furthermore, we demonstrate that the electrochemical performance of the PC NP layers can be significantly improved by the periodic insertion of metal NP layers between the PC NP layers, in which metal NP layers can effectively prevent any substantial decrease in the electrical conductivity of PC NP layers. The assembled MP-SC electrode exhibit remarkable areal power (15.1 mW cm$^{-2}$, specific power density: ~128.9 kW kg$^{-1}$), energy density (267.3 μW h cm$^{-2}$, specific energy density: ~121.5 W h kg$^{-1}$), and areal capacitance (1.35 mF cm$^{-2}$) values at a high loading mass of PC NPs (>4.09 mg cm$^{-2}$) with a long cycling operation (~90% of initial capacitance after 5,000 cycles). Considering that our approach can easily impart metal properties to various insulating substrates (i.e., polyester, nylon, poly(ethylene terephthalate), and cellulose papers), we believe that our approach can be widely used for various electrodes, including energy storage electrodes for portable/wearable electronics.

## Results

### Preparation and physical characterization of (TREN/TOABr-Au NP)$_n$ multilayers.
To prepare MP electrodes that contained a large amount of metal and PC NPs that preserved the highly porous structure of the cellulose papers (Korean traditional paper "Hanji") (Fig. 1a), we first investigated the adsorption behavior and electrical properties of the TREN/TOABr-Au NP multilayer films. For this study, TOABr-Au NPs dispersed in toluene were sequentially LbL-assembled with TREN in ethanol via a ligand replacement reaction between bulky TOABr and TREN (Supplementary Fig. 1). In this case, the Au NP arrays buried within the (TREN/TOABr-Au NP)$_n$ multilayers scarcely contained the bulky TOABr ligands, and additionally, only one molecule layer (i.e., TREN, with M$_w$ ~146) existed between the vertically adjacent Au NP layers. This approach differs from conventional NP film methods that require two or three organic layers between adjacent NP layers (Fig. 1b). Additionally, the LbL assembly of Au NPs in organic media significantly increases the loading mass of Au NPs per layer in the lateral dimension because there is no electrostatic repulsion among the neighboring NPs. Furthermore, these Au NP multilayers exhibited the dense and disordered structure with a large amount of nanopores formed among Au NPs as well as the macropores of the cellulose paper. As a result, it was considered that this structural uniqueness of MP electrodes could facilitate the ion transfer, which is closely related to the power performance of MP-SC. These phenomena were confirmed by ultraviolet–visible (UV–Vis) spectroscopy, cross-sectional field-emission scanning electron microscopy (FE-SEM), atomic force microscopy (AFM), thermogravimetric analyses, and quartz crystal microbalance (QCM) measurements (Supplementary Figs. 2–4).

Based on these results, we examined the electrical properties of the (TREN/TOABr-Au NP)$_n$ multilayers coated onto quartz glass as a function of the bilayer number ($n$) using the four-probe measurement method (Fig. 2a). As the bilayer number ($n$) increased from 5.5 ($36 \pm 2$ nm thickness) to 30.5 ($215 \pm 4$ nm thickness), the sheet resistance of the multilayers sharply decreased from 10.3 to $0.23\ \Omega$ sq$^{-1}$; moreover, their electrical conductivity ($\sigma$) increased from $2.7 \times 10^4$ to $2.1 \times 10^5$ S cm$^{-1}$

(the conductivity of the bulk gold: ~$4.1 \times 10^5$ S cm$^{-1}$ at 20 °C) without any thermal, mechanical, and/or percolation treatment. This electrical conductivity ($2.1 \times 10^5$ S cm$^{-1}$) is significantly higher than those of the previous reported flexible conductors ($1–10^4$ S cm$^{-1}$)[23, 24, 37, 45–47]. In contrast, the LbL-assembled (poly(ethylene imine) (PEI, M$_w$ ~1,500)/TOABr-Au NP)$_{30.5}$ films using amine-functionalized polymers with a relatively high molecular weight (instead of TREN) exhibited a relatively high sheet resistance of $90\ \Omega$ sq$^{-1}$ and a low conductivity ($\sigma$) of $6.7 \times 10^2$ S cm$^{-1}$, showing the strong electrical dependence on the type of the organic layer bridging the Au NPs (Supplementary Fig. 5). Additionally, when the outermost ligands of the nanocomposite films were changed from bulky TOABr to TREN, the sheet resistance was more significantly decreased relative to that obtained after changing the outermost ligands from TOABr to PEI (Fig. 2b). However, the (anionic poly(styrene sulfonate) (PSS)/cationic dimethylaminopyridine (DMAP)-Au NP)$_{30.5}$ multilayers prepared from the electrostatic LbL assembly in aqueous media exhibited an insulating behavior (>$10^9\ \Omega$ sq$^{-1}$) irrespective of the outermost layer. These results indicate that the bulky organic ligands and polymer linkers that were bound to the surface of the metal NPs interrupted the electron transport among the neighboring metal NPs.

To investigate the electron transfer mechanism of the (TREN/TOABr-Au NP)$_n$ multilayers, the temperature-dependent electrical conductivity was monitored by four-probe measurement over a temperature range of 2–300 K (Supplementary Fig. 6). In this case, the temperature dependence of conductivity did not correspond to the linear dependence with the following equation for semiconducting kinetics: $\sigma = \sigma_0 \exp(-A/T^{(1/d+1)})$ for variable-range hopping ($d$=3) or tunneling ($d$=1) mechanism, where $\sigma$ is the conductivity, $T$ is the absolute temperature (K), $A$ is a constant, and $d$ is the dimensionality[37, 48]. As a result, the electron transport mechanism of the (TREN/TOABr-Au NP)$_n$ multilayers is based on the metallic conduction behavior originating from the percolated networks of Au NPs rather than hopping or tunneling conduction behavior. As the temperature was decreased from 300 to 2 K, the electrical resistivity of the (TREN/TOABr-Au NP)$_{30.5}$ multilayers gradually decreased, which represented typical metallic behavior[49]. In this case, the formed Au NP multilayers showed a positive temperature coefficient of $1.64 \times 10^{-3}$ K$^{-1}$ obtained by following relationship: $\Delta R/R_0 = \alpha \Delta T$, where $R$ and $\alpha$ are the resistance ($\Omega$) and the temperature coefficient, respectively. Notably, the high electrical conductivity of the (TREN/TOABr-Au NP)$_n$ multilayers could be maintained under high mechanical stress/strain. To demonstrate such capability, the (TREN/TOABr-Au NP)$_{15.5}$ multilayers were deposited onto poly(ethylene terephthalate) (PET) films that had a high affinity for amine groups; then, the conductivity change ($\sigma/\sigma_0$) of the (TREN/TOABr-Au NP)$_{15.5}$-coated 105 nm thick PET films, with a conductivity of $7.3 \times 10^4$ S cm$^{-1}$ (sheet resistance: ~$1.3\ \Omega$ sq$^{-1}$) at the initial (flat) state ($\sigma_0$), was investigated as a function of the bending radius and cycling number (Fig. 2c and Supplementary Fig. 7). In this case, the conductive multilayers displayed excellent electrical stability by retaining 96% of the initial conductivity, even after 10,000 bending cycles (bending radius=1.5 mm). These phenomena were in stark contrast to the evaporated Au-coated films (i.e., 100 nm thick Au/Ti/PET), which lost ~92% of their initial conductivity value after the same number of cycles. As a result, it is concluded that the Au NP-coated films can mitigate the impact of external mechanical stimuli by minimizing the contact area with the substrates.

Furthermore, the formed multilayer films also exhibited high stability in a variety of solvents, from organic media to

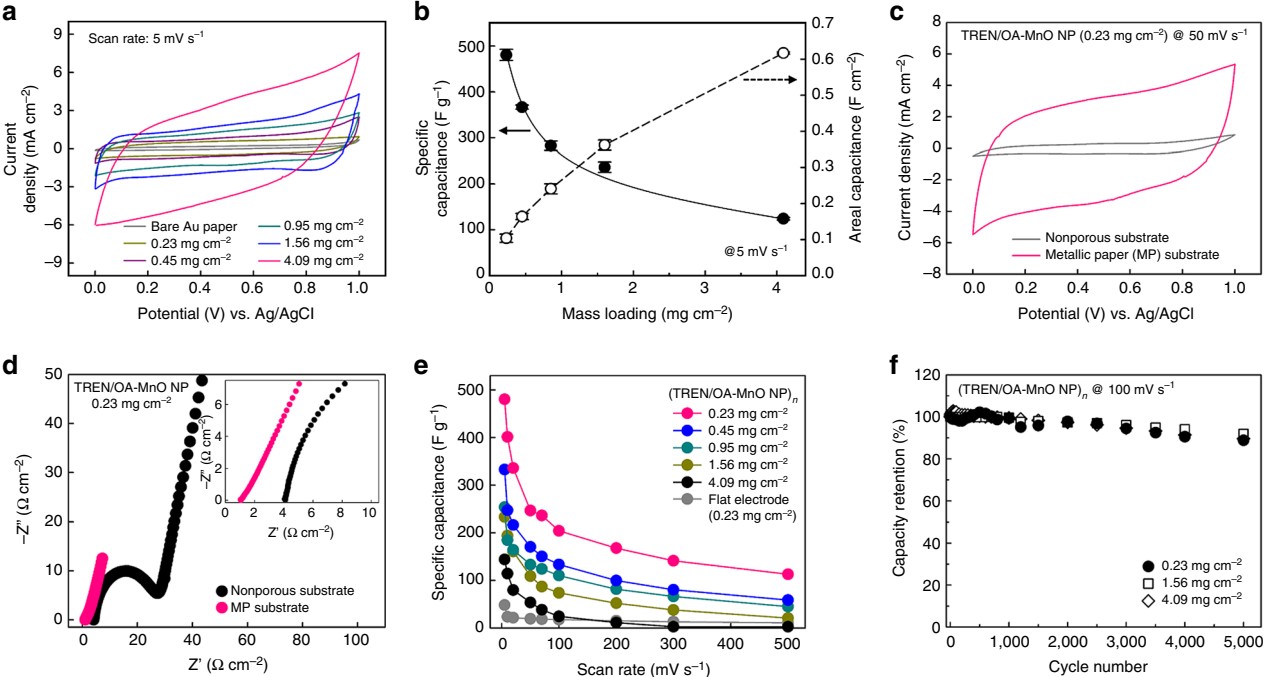

**Fig. 3** Electrochemical properties of the MP-based (TREN/OA-MnO NP)$_n$ electrode. **a** Cyclic voltammograms (CVs) of the (TREN/OA-MnO NP)$_n$-coated MP electrodes as a function of the mass loading of MnO NPs (mg cm$^{-2}$) at a scan rate of 5 mV s$^{-1}$. **b** Specific and areal capacitances of the (TREN/OA-MnO NP)$_{20.5}$-coated MP electrodes as a function of the mass loading of MnO NPs (from 0.23 to 4.09 mg cm$^{-2}$). **c** CVs of the (TREN/OA-MnO NP)$_n$-coated MP and nonporous substrate (Au-coated Si wafer) electrodes with a mass density of 0.23 mg cm$^{-2}$ at a scan rate of 50 mV s$^{-1}$. **d** Nyquist plots of the (TREN/OA-MnO NP)$_n$-coated MP and nonporous substrate nonporous substrate (Au-coated Si wafer) electrodes. **e** Scan rate dependence of the specific capacitance of the (TREN/OA-MnO NP)$_n$-coated MP electrodes with different mass densities. **f** Cycling retention of the (TREN/OA-MnO NP)$_n$-coated MP electrodes with mass densities of 0.23, 1.56 and 4.09 mg cm$^{-2}$, respectively

pH-controlled aqueous media, due to covalent bonding between the primary amine groups and Au NPs (Supplementary Fig. 8). We also highlight that our approach can be widely applied to a variety of other substrates, such as conventional papers, polyester textiles, and nylon threads, due to the high affinity for the amine groups of TREN and, as a result, can convert the insulating substrates into fully flexible metallic conductors without any substantial change in the native porous structure (Supplementary Figs. 9 and 10).

**Preparation of cellulose paper-based electrode.** In particular, in the case of 140 µm-thick cellulose papers with highly porous structures, it was confirmed by FE-SEM and energy-dispersive X-ray spectroscopy (EDX) that their interior and exterior were uniformly and densely coated by the (TREN/TOABr-Au NP)$_{10.5}$ multilayers (Fig. 2d). As shown in Fig. 2d, the (TREN/TOABr-Au NP)$_{10.5}$ multilayer-coated papers maintained their high porous three-dimensional (3D) structures. Additionally, the elemental mapping images of the cross-sectional MP clearly demonstrated that the Au NPs were homogeneously deposited within the interior of the paper without agglomeration (right in Fig. 2d). Although the total mass density of the formed MPs was measured to be ~0.36 g cm$^{-3}$ (the mass density of the pristine paper: ~0.29 g cm$^{-3}$), the loading amount of the adsorbed Au NPs per layer on the porous paper was 27 times higher than that of the Au NPs on the nonporous flat substrates, such as the QCM electrode. The nanocomposite papers, while lightweight, exhibited an extremely low sheet resistance of 0.31 Ω sq$^{-1}$ and a high electrical conductivity of 230.1 S cm$^{-1}$ (i.e., conductivity of the total films, including the 140 µm-thick paper) without thermal treatment or mechanical pressing (Fig. 2e). As shown in Fig. 2f, by increasing the bilayer number ($n$) of the (TREN/

TOABr-Au NP)$_n$ multilayers from 0 to 10.5, the insulating paper (i.e., $n=0$) was gradually converted into the MP, showing a color change from white to yellow. Interestingly, a light-emitting diode (LED) connected to the as-prepared MP conductors could be activated even after the deposition of only one Au NP layer (i.e., TREN/TOABr-Au NP/TREN), implying the dense loading of the Au NPs with reduced interparticle distances on the porous 3D cellulose networks. Furthermore, our approach could be easily applied to large-area paper, and the paper exhibited excellent electrical stability under severe mechanical deformation (Fig. 2f). With additional heating and/or mechanical pressing, the electrical conductivity was further enhanced (Supplementary Fig. 11). It should also be noted that these metallic papers cannot be prepared by previously reported approaches such as thiol chemistry or conventional electrostatic LbL assembly[27–40].

**Electrochemical properties of MP-SCs.** Based on these results, we explored the possibility of effectively using the MPs as supercapacitor electrodes. First, the OA-MnO NPs as high-energy PC materials were successively deposited onto the MPs (i.e., (TREN/TOABr-Au NP)$_{10.5}$ multilayer-coated papers) using the above-mentioned ligand replacement reaction (i.e., a higher affinity for the metal oxide with the amine groups of TREN than the OA ligands). In this case, TREN could directly bridge the interfaces of the paper/metal/metal oxide NPs, resulting in the highly integrated porous 3D networks (Supplementary Fig. 12). The mass loading of the (TREN/OA-MnO NP)$_n$-coated MPs (MnO-MP) was found to increase linearly with the bilayer number, indicating the precise controllability of the capacity (Fig. 2h). In particular, the mass loading of the MnO NP multilayers adsorbed on the MPs was ~10 times higher than that of the multilayers on the nonporous

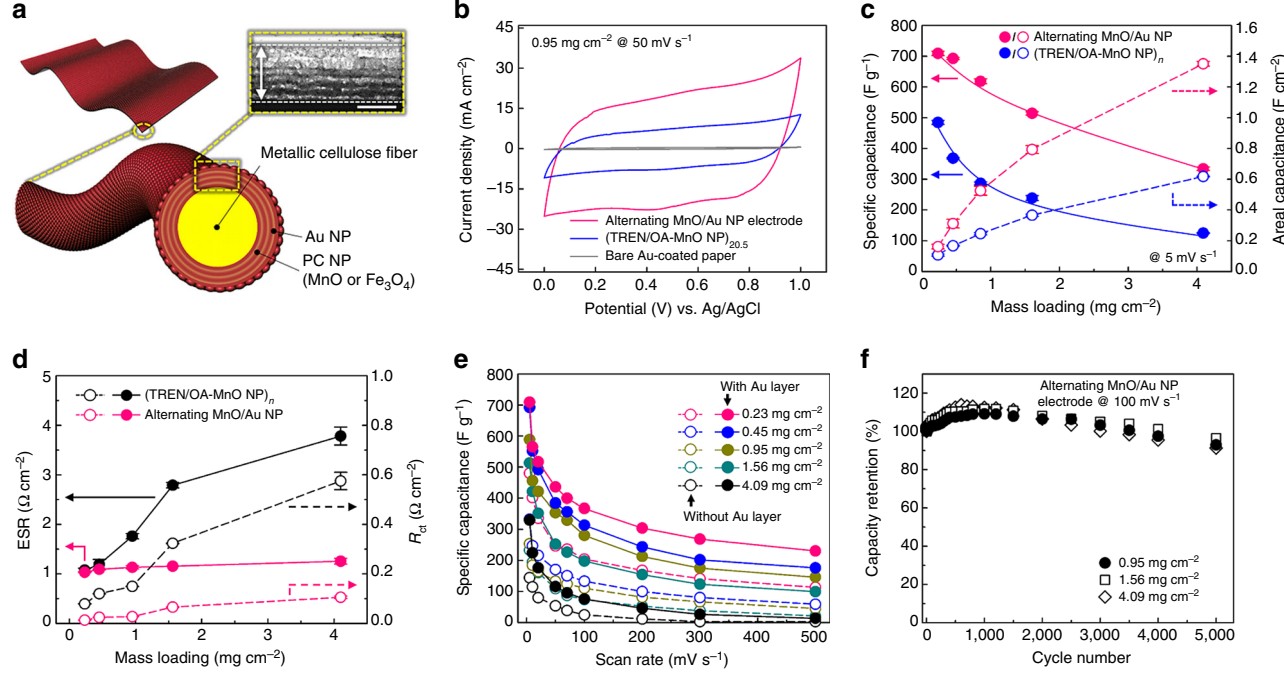

**Fig. 4** Electrochemical properties of the alternating MnO/Au NP electrode. **a** Schematics of the alternating PC/Au NP electrodes. The FE-SEM micrograph shows the cross-sectional view of the alternating MnO/Au NP electrode. The size of the *scale bar* is 200 nm. **b** Comparison of the current responses of the alternating MnO/Au NP and the (TREN/OA-MnO NP)$_{20.5}$-coated MP electrodes at a scan rate of 50 mV s$^{-1}$. **c** Specific and areal capacitances of alternating MnO/Au NP electrodes as a function of the mass loading of MnO NPs at a scan rate of 5 mV s$^{-1}$. **d** Traces of the ESR and $R_{ct}$ values of the alternating MnO/Au NP and (TREN/OA-MnO NP)$_n$ electrodes as a function of the mass loading, respectively. All values were collected from the Nyquist plot of each electrode in Supplementary Figs. 14 and 20. **e** Specific capacitances of the alternating MnO/Au NP (with Au layer) and the (TREN/OA-MnO NP)$_n$ (without Au layer)-coated MP electrodes with different mass densities as a function of the scan rate, respectively. **f** Cycling retention of alternating MnO/Au NP electrodes with mass densities of 0.95, 1.56 and 4.09 mg cm$^{-2}$, respectively

metallic substrates, which was expected to significantly improve the areal capacity of the electrodes.

The intrinsic electrochemical properties of the MnO-MP-SC electrodes were evaluated using a three-electrode cell configuration in a 1 M Na$_2$SO$_4$ electrolyte. Cyclic voltammetry (CV) scans of the electrodes at a scan rate of 5 mV s$^{-1}$ were measured (Fig. 3a), and the corresponding specific and areal capacitances were evaluated as a function of the mass loading of the MnO NPs (mg cm$^{-2}$) (Fig. 3b). With an increased loading amount (or bilayer number, $n$) of the MnO NPs within the multilayers, the current levels of the CV curves and corresponding areal capacitances were enhanced with the increase of the integrated area of the CVs. In particular, even MnO-MP-SC electrodes containing a high mass loading of 4.09 mg cm$^{-2}$ exhibited excellent charge storage performance with quasi-rectangular CV curves, allowing fast reversible multielectron redox reactions of the MnO NPs[50]. It should be noted that the excellent charge storage behavior of the MnO-MP-SC electrodes with high mass loading can be achieved not only by structural benefits of the highly porous cellulose paper, but also by effective removal of insulating bulky ligands (i.e., TOABr or OA) bound to the surface of NPs during ligand-mediated LbL assembly process. That is, the ligand exchange from electrochemically nonactive and bulky ligands to hydrophilic TREN molecules allows the low charge transfer resistance, better penetration of the electrolyte, and enhanced utilization of active materials. Galvanostatic charge/discharge (GCD) curves with triangular features were also indicative of the favorable capacitive behavior of the MnO-MP-SC electrodes (Supplementary Fig. 13). The maximum areal capacitance of the electrode was 617 mF cm$^{-2}$ at the high loading density of 4.09 mg cm$^{-2}$. Although this value is higher than or

comparable to those reported in previous papers based on textile electrodes (Supplementary Table 1), the areal capacitance of the electrodes and the loading amount of MnO NPs in our system can be further scaled by simply increasing the bilayer number. In contrast, the specific mass capacitance of the MnO-MP-SC electrodes decreased with increasing loading of the MnO NPs due to the electronic and ionic resistances of MnO NPs with low electrical conductivity values and low ion diffusion coefficients, which limits their PC performance[21, 51]. Electrochemical impedance spectroscopy (EIS) measurement of the MnO-MP-SC electrodes with different mass densities further revealed similar loading-dependent trends, indicating an increase in the equivalent series resistance (ESR) with increasing loading density (Supplementary Fig. 13).

Notably, the high specific capacitances of the MP-SC electrodes could not be easily achieved using other conventional electrodes. Figure 3c shows CV scans of (TREN/OA−MnO NP)$_n$-coated MP and the Au-evaporated Si-wafer electrodes at the same mass density (i.e., ~0.23 mg cm$^{-2}$). The current response of the MP-SC electrode was 10 times higher than that of the nonporous SC electrode; consequently, the MP-based electrodes exhibited a high specific mass capacitance of 481 F g$^{-1}$ at a scan rate of 5 mV s$^{-1}$, in contrast to 48 F g$^{-1}$ for the nonporous electrodes. The EIS further confirmed the facilitated kinetics of the MP-SC with the significantly decreased charge transfer resistance of 0.08 Ω cm$^{-2}$ relative to that of the nonporous SC electrodes (23.4 Ω cm$^{-2}$) (Fig. 3d and Supplementary Fig. 14). The enhanced electrochemical performance of the MP-SC electrodes could be attributed to their highly porous structure with a large surface area, which decreased the transport length scale of the ions and electrons[52]. Moreover, the MnO-MP electrodes exhibited favorable PC behavior as the scan rate was increased from 5

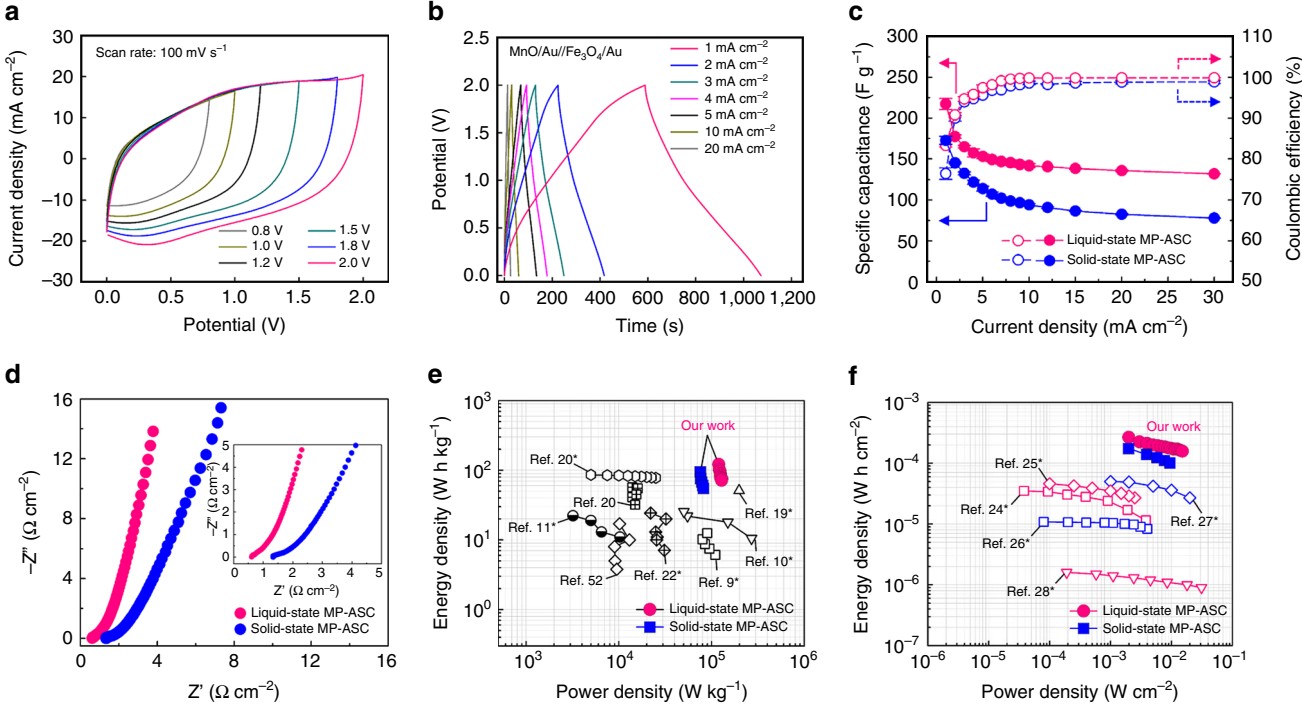

**Fig. 5** Electrochemical properties of the MP-asymmetric supercapacitor (ASC). **a** CVs of the liquid-state MP-ASC (i.e., MnO/Au//Fe$_3$O$_4$/Au in 1 M Na$_2$SO$_4$) at a scan rate of 100 mV s$^{-1}$. **b** Galvanostatic charge/discharge (GCD) curves of the liquid-state MP-ASC at various current densities in the range of 1–10 mA cm$^{-2}$. **c** Specific capacitance and coulombic efficiency of the MP-ASC as a function of the current density. **d** Nyquist plots of the MP-ASCs. The *inset* shows a higher magnification of the high-frequency region. Ragone plots of the solid- and liquid-state MP-ASCs as a function of **e** the total mass of the active energy materials and **f** the area of the MP-ASCs compared with previously reported flexible asymmetric supercapacitors (*Supplementary References)

to 500 mV s$^{-1}$ (Supplementary Fig. 15). The specific mass capacitance values obtained from the CVs gradually decreased as the scan rate increased, which was normal behavior for PC material-based charge storage systems (Fig. 3e). Although the Au and MnO NP layers were densely coated on the cellulose fibers within the papers, the hydrophobic TOABr (for Au NPs) and OA ligands (for MnO NPs) bound to the surface of the NPs were replaced by TREN, and consequently the NPs buried between the TREN layers were changed to the hydrophilic NPs. Therefore, the NP-coated cellulose fibers within the highly porous MP-SC electrodes could act as favorable electrolyte reservoirs and provide an effective pathway for ion transport due to the water-swelling effect of cellulose fibers in aqueous media[53]. Furthermore, we also highlight that the TREN, as a small-molecule linker with amine moieties, supported not only facile charge transport due to the significant decrease of the interdistance between the NPs but also stable covalent bonds between adjacent NPs. The MnO-MP electrodes with 0.23, 1.56 and 4.09 mg cm$^{-2}$ maintained 88.9, 91.8, and 89.6% of their initial capacitances, respectively, after 5,000 cycles during CV at 100 mV s$^{-1}$ (Fig. 3f).

**Improving the electrochemical performance of MP-SCs by LbL design.** Despite their high loading density and areal capacitance values, the gradual decrease in specific capacitances and the rate capability with a high loading density is a critical drawback of the PC NP-based pseudocapacitors. To resolve these problems and further enhance the performance, the metal NP layers were periodically inserted into the (TREN/OA-MnO NP)$_n$ multilayers, as shown in Fig. 4a (i.e., MP/[(TREN/OA-MnO NP)$_3$/(TREN/TOABr-Au NP)$_1$]$_m$ or the alternating MnO/Au NP

electrodes). In this case, the inserted Au NPs were uniformly distributed within the center inside of the electrode and could provide favorable electron transfer pathways between adjacent MnO NP layers (Supplementary Figs. 16 and 17). Figure 4b shows the CVs of the alternating MnO/Au NP and (TREN/OA-MnO NP)$_{20.5}$ electrodes without the inserted Au NP layers and bare MP electrodes at a scan rate of 50 mV s$^{-1}$. Interestingly, the electrochemical response of the alternating MnO/Au NP electrodes was superior to that of the (TREN/OA-MnO)$_{20.5}$-coated MP electrode without the Au NP layers despite the same mass density of 0.95 mg cm$^{-2}$, which was consistent with the GCD behavior (Supplementary Fig. 18). The excellent response of the alternating MnO/Au NP electrodes could also be observed in the mass-dependent CVs acquired at a scan rate of 5 mV s$^{-1}$ (Supplementary Fig. 19), which displayed larger and more rectangular features than those of the (TREN/OA-MnO NP)$_n$ electrodes without Au NP layers, as shown in Fig. 3a. As a result, the alternating MnO/Au NP electrodes exhibited specific and areal capacitance values of 709 F g$^{-1}$ (at 0.23 mg cm$^{-2}$) and 1.35 F cm$^{-2}$ (at 4.09 mg cm$^{-2}$) at a scan rate of 5 mV s$^{-1}$, respectively (Fig. 4c). This specific capacitance value exceeds those of previously reported papers on textile-type SC electrodes (Supplementary Table 1). Furthermore, the areal capacitance can be further improved by increasing the periodic number (*m*) of nanocomposite electrodes. Given that the double-layer capacitance of the inserted Au NP layer was smaller than that of the PC MnO NPs, this dramatic increase in the electrochemical response implied that the periodically inserted metal layers significantly reduced the internal resistance, as confirmed by the EIS measurement (Fig. 4d and Supplementary Fig. 20). The ESR and charge transfer resistance ($R_{ct}$) values of the alternating

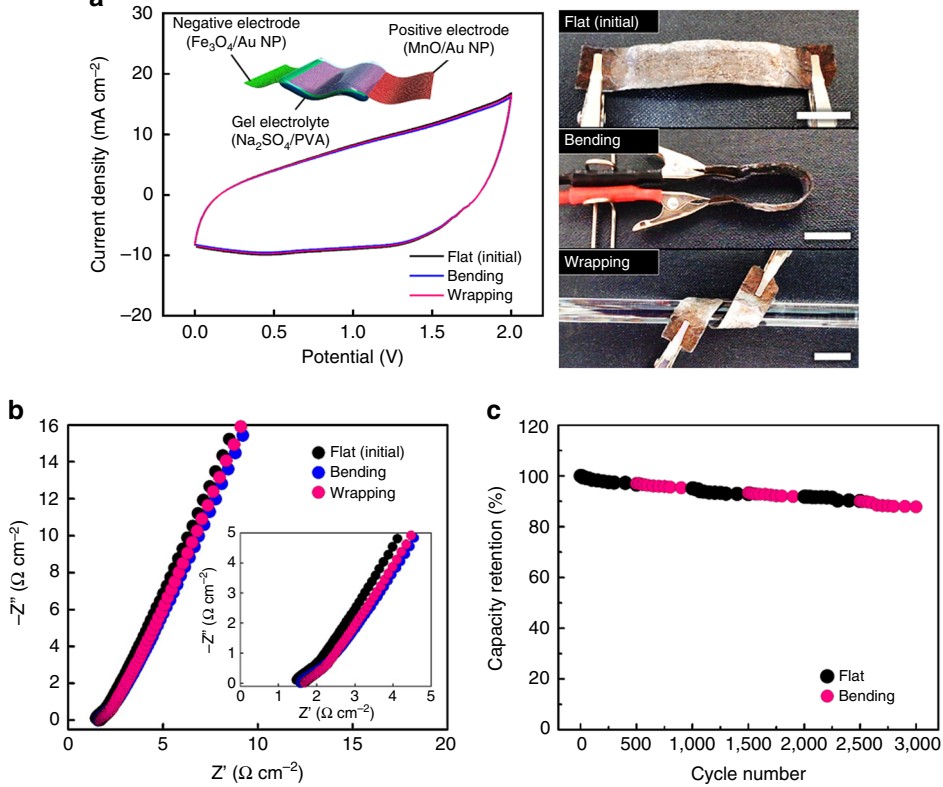

**Fig. 6** Mechanical properties of the MP-ASCs. **a** CVs of the solid-state MP-ASCs at a scan rate of 50 mV s$^{-1}$ recorded under flat (initial), bending and wrapping conditions. The *scale bars* in the photographs (*right*) correspond to 1 cm. **b** Nyquist plot of the solid-state MP-ASCs with different flexible conditions. **c** Cycle retention test of the solid-state MP-ASCs with different bending conditions at a scan rate of 100 mV s$^{-1}$

MnO/Au NP electrodes were nearly constant and slightly increased, respectively, with increasing loading mass density, while those of the (TREN/OA-MnO NP)$_n$ electrodes were significantly increased (Fig. 4d). Additionally, the alternating MnO/Au NP electrodes exhibited improved capacitive behaviors compared to the (TREN/OA-MnO NP)$_n$ electrodes at the scan rate of 5–500 mV s$^{-1}$ (Supplementary Fig. 21). These results indicate that the charge transfer properties of the alternating MnO/Au NP electrodes are superior to those of the (TREN/OA-MnO NP)$_n$ multilayer electrodes. Figure 4e shows the specific capacitances of the alternating MnO/Au NP and (TREN/OA-MnO NP)$_n$ electrodes at a scan rate ranging from 5 to 500 mV s$^{-1}$. Although the specific capacitance of the alternating MnO/Au NP electrodes gradually decreased with the increasing scan rate due to the limited diffusion properties of the PC NPs, these electrodes still exhibited a higher specific capacitance than that of the electrodes without Au NP layers over the entire scan rate range. In particular, in the case of the alternating electrode with an areal mass density of 0.23 mg cm$^{-2}$, its specific capacitance was measured to be ~230 F g$^{-1}$ even at a fast scan rate of 500 mV s$^{-1}$, displaying an excellent rate capability. The more effective charge transport phenomena of the alternating MnO/Au NP electrodes can be further clarified by the loading-dependent electrochemical behavior with different scan rates (Supplementary Fig. 22). Furthermore, the alternating MnO/Au NP electrodes with mass densities of 0.95, 1.56 and 4.09 mg cm$^{-2}$ maintained 93, 96.4 and 91.3% of their initial capacitance after 5,000 cycles, respectively (Fig. 4f).

**Electrochemical and mechanical performance of MP-ACSs.** Based on these results, we prepared hybrid asymmetric supercapacitors (ASCs) composed of alternating MnO/Au NP

(positive electrode) and alternating Fe$_3$O$_4$/Au NP electrode (negative electrode). The preparation method, structural composition, and electrochemical properties of the Fe$_3$O$_4$ NP-based negative electrodes were similar to those of the MnO NP-based electrodes (see Methods and Supplementary Figs. 23–25). In this case, the voltage window was increased up to 2 V in liquid (1 M Na$_2$SO$_4$) or solid (Na$_2$SO$_4$/poly(vinyl alcohol) (PVA)) electrolytes. The MP-ASCs measured at a scan rate of 100 mV s$^{-1}$ exhibited potential-dependent quasi-rectangular-shaped CV curves with typical capacitive characteristics, suggesting the fast Faradaic reaction of the PC NPs within the highly porous MP electrodes (Fig. 5a and Supplementary Fig. 26). This stable storage behavior was also observed during GCD measurements (Supplementary Fig. 27). The symmetric triangular GCD curves and quasi-rectangular CVs of the MP-ASCs at different sweep rates indicate desirable capacitive features and excellent rate capabilities (Fig. 5b and Supplementary 28). Figure 5c shows the specific capacitance and coulombic efficiency of the MP-ASCs calculated from GCD profiles with different current densities of 1–10 mA cm$^{-2}$. The maximum specific capacitances based on the total active materials containing both positive and negative electrodes (3.1 mg) was measured to be 222 F g$^{-1}$ for the liquid-state MP-ASC and 176 F g$^{-1}$ for the solid-state MP-ASC at a current density of 1 mA cm$^{-2}$ (0.65 A g$^{-1}$), which were much more favorable values than those of previously reported paper or textile-based ASCs (including cellulose paper, cotton, polyester, and carbonaceous fabric substrates)[54–57]. The MP-ASCs also exhibited a satisfactory capacity retention of 138 F g$^{-1}$ for the liquid-state MP-ASC and 80 F g$^{-1}$ for the solid-state MP-ASC at a high current density of 20 mA cm$^{-2}$. Although the solid-state MP-ASC in the present study showed a lower capability than the liquid-state device due to the poor ion diffusion nature of

polymer gel electrolyte, this parameter could be improved by chemically modifying the gel electrolyte[58]. Additionally, the assembled MP-ASCs exhibited an excellent coulombic efficiency of over 90% at a current density of 2 mA cm$^{-2}$ and reached 99.8% for the liquid-state MP-ASC and 99% for the solid-state ASC at a current density of 20 mA cm$^{-2}$. The EIS of the MP-ASCs also reflected efficient ion migration characteristics during the electrochemical sweep (Fig. 5d). The ESR values at a high frequency (100 kHz) were measured to be 0.61 and 1.34 Ω cm$^{-2}$ for the liquid- and solid-state MP-ASCs, respectively, indicating the notably low total internal resistance of the devices.

The power and energy densities of the MP-ASCs were evaluated for the total mass of the active materials and area of the devices from galvanostatic discharge profiles (Figs. 5e, f and Supplementary Fig. 29). Figure 5e shows the specific power and energy densities of the solid- and liquid-state MP-ASCs at various current densities along with the corresponding values for liquid-state flexible SCs reported by other research groups. In this case, the MP-ASCs exhibited maximum power and energy densities of 128.9 kW kg$^{-1}$ and 121.5 W h kg$^{-1}$ for the liquid state and 95.1 kW kg$^{-1}$ and 81.7 W h kg$^{-1}$ for the solid state, respectively, even at high mass loadings of the active energy materials, which were superior to the other previously reported flexible SCs (Supplementary Table 2). These remarkable electrochemical outputs for both the power and energy densities of the MP-ASCs suggest that the porous 3D networks of the cellulose paper and periodically inserted Au NPs played a synergistic role in reducing the internal resistance caused by dense packing of the PC NPs. The areal performance is considered a valuable factor for evaluating flexible and wearable two-dimensional (2D) textile energy storage electrodes[2]. As shown in Fig. 5f, the maximum areal power and energy densities of the MP-ASCs were measured to be 15.1 mW cm$^{-2}$ and 267.3 µW h cm$^{-2}$ for the liquid state and 9.5 mW cm$^{-2}$ and 174.7 µW h cm$^{-2}$ for the solid state, respectively. To our knowledge, this performance is significantly better than that of previously reported flexible SCs and can be easily improved further by increasing the bilayer number. The long-term stability of the MP-ASCs was tested by GCD cycles at a current density of 20 mA cm$^{-2}$ for up to 5,000 cycles (Supplementary Fig. 30). The MP-ASCs showed an excellent capacity retention of 89.8% for the liquid-state MP-ASC and 89.6% for the solid-state MP-ASC of their initial values.

The MP-ASCs also displayed excellent mechanical stability under various stress conditions, which is one of the essential factors for real applications. Figure 6a shows the CVs of the solid-state MP-ASC under bending and wrapping. In this case, there was no meaningful change in the CV geometries, indicating the robust and stable connections among the cellulose paper substrate, NPs, and electrolyte. This mechanical stability was also confirmed by monitoring the ESR changes in the EIS (Fig. 6b). The changes in the ESR values were negligible, which indicated the excellent flexibility of the MP-ASC. The fatigue resistance of the MP-ASC was further tested by continuous CV cycling with different bending angles (Fig. 6c). As shown in Fig. 6c, the capacity retention was nearly constant for 3,000 cycles under sequential conversion of the bending conditions. These results highlight that our devices can be potentially applied to flexible and portable power applications with high performance.

## Discussion

We demonstrated that the ligand-mediated LbL-assembly of metal NPs could convert the cellulose papers to highly porous/flexible current collectors; additionally, these collector films could significantly increase the loading amount of

high-energy PC NPs in both lateral and vertical dimensions (due to the porous structure of the papers and LbL assembly). The flexible SC electrodes based on the metal NP/PC NP-incorporated papers exhibited remarkable power/energy densities and scalable areal capacitances. In particular, we highlight that the alternate adsorption of metal and PC NPs onto the MPs substantially increased the energy storage capabilities and maintained high electrical conductivity, despite the thickness increase of the poorly conductive PC NP layers. Given that our approach can be effectively applied to various conductive and active materials irrespective of substrate size and shape, we suggest that our approach can provide a facile and versatile basis for designing high-performance electrodes that require a large surface area, flexibility similar to paper or textiles, and precise control of the thickness or loading density. We also envision that this approach can be extended to the biomass-derived carbon materials[59, 60] with a highly porous architecture for developing large-scale energy storage devices with high performance.

## Methods

**Synthesis of hydrophobic NPs**. *TOABr-Au NP*: Au NPs protected TOABr (Sigma-Aldrich) were synthesized using a two-phase reaction reported as the Brust method[61]. Briefly, 30 mM of gold(III) chloride trihydrate (HAuCl$_4$·3H$_2$O, ≥99.9%, Sigma-Aldrich) in deionized water (30 ml) and 20 mM of TOABr-dispersed toluene (80 ml) were mixed with vigorous stirring. Then, a 0.4 M aqueous solution of NaBH$_4$ (25 ml, 99.99%, Sigma-Aldrich) was added to the above mixture for reduction. After that, the toluene phase was separated from the aqueous solution and washed with H$_2$SO$_4$ (0.1 M, 95% purity, Daejung Chemicals), NaOH (0.1 M, 97%, Sigma-Aldrich), and deionized water several times.

*OA-MnO NP*: MnO NPs measuring ~11 nm in toluene were synthesized as previously reported[62]. In a typical experiment, a Mn-oleate precursor complex was first prepared by the thermal reaction of a mixture of manganese(II) chloride tetrahydrate (MnCl$_2$·4H$_2$O, ≥98%, Sigma-Aldrich), sodium oleate (>97%, TCI Co., Ltd), ethanol, *n*-hexane, and deionized water at 70 °C for 12 h, followed by eliminating the solvent. The obtained pink-colored Mn-oleate powder (1.24 g) was mixed with 1-hexadecene (12.8 ml) at 70 °C for 1 h, then gradually heated to 280 °C at a rate of 2 °C min$^{-1}$ under argon conditions and maintained for 10 min. The resultant mixture was cooled and washed by centrifugation. The hydrophobic OA-MnO NPs were redispersed in toluene or hexane.

*OA-Fe$_3$O$_4$ NP*: Fe$_3$O$_4$ NPs with a diameter of 8 nm in toluene were prepared using the thermal reaction of Fe(acac)$_3$ (2 mmol, ≥99.9%, Sigma-Aldrich), 1,2-hexadecanediol (10 mmol, 90%, Sigma-Aldrich), oleylamine (6 mmol), oleic acid (5 mmol), and benzyl ether (20 ml) under N$_2$ conditions[63]. Briefly, the mixture was first heated to 200 °C for 2 h, after which the temperature was increased to 300 °C for 1 h for refluxing. The cooled resultant solution was separated into NPs and excess residual solvents by centrifugation with ethanol. Additional oleic acid and oleylamine were added to the product solution and purified using centrifugation (6,000 r.p.m., 10 min) several times. The obtained black-brown NP powder was dispersed in toluene (or hexane).

*Preparation of the MP electrode*: The hydrophobic NPs (including Au, MnO, and Fe$_3$O$_4$) in toluene (5 mg ml$^{-1}$) and TREN in ethanol (1 mg ml$^{-1}$) were deposited alternately onto a commercial paper substrate (Korean Hanji (mulberry) paper) with a size of 1 cm × 2 cm) using LbL assembly with a dipping time of 30 min. First, a paper substrate was dipped into the TREN solution for 30 min, then washed with pure ethanol to remove the weakly adsorbed TREN molecules. Subsequently, the TREN-coated paper substrate was immersed into the hydrophobic NP solution for 30 min and then washed with pure toluene, resulting in one bilayer film (referred to as the (TREN/hydrophobic NP)$_1$ film). These procedures were repeated to obtain electrodes with the desired thickness (or mass density). Additionally, the above-mentioned deposition processes can be applied in the same manner to various substrates, such as silicon wafers, QCM electrodes, PET films, polyester textiles, or nylon threads.

*Characterization*: Fourier transform infrared spectra of the multilayers were collected using a CARY 600 spectrometer (Agilent Technology) in specular mode with a resolution of 4 cm$^{-1}$, and the obtained data were plotted using spectrum analysis software (OMNIC, Nicolet). FE-SEM and EDX were conducted using an S-4800 (Hitachi). Transmission electron microscopy of synthesized NPs was conducted using a Tecnai20 (FEI). UV–Vis spectra of the LbL-assembled multilayer films onto quartz glass slides were recorded using a Lambda 35 (Perkin Elmer) within a scan range of 200–800 nm. The surface morphology of the formed multilayers was scanned using AFM (XE-100, Park Systems) in tapping mode. The film growth was quantitatively monitored using a QCM (QCM 200, SRS). In this case, the mass change of each layer was calculated from the QCM frequency change using the Sauerbrey equation[64]. The temperature dependence of the electrical conductivity of the formed MPs was measured using a physical property measurement system (PPMS-9, Quantum Design) over a temperature range from 2

to 300 K. Error bars in electrical properties and mass loading of the multilayers represent the s.d. of the each values obtained from three samples.

**Preparation of the ASCs**. During the assembly of the MP-ASCs, alternating MnO/Au NP and Fe₃O₄/Au NP electrodes were employed as the positive and negative electrodes, respectively. In this case, the charge balance (q⁺=q⁻) of each electrode was adjusted by controlling the loading amount of the PC NPs using the following relationship[65]:

$$m^+/m^- = C^- \Delta E^-/C^+ \Delta E^+ (q = C \times \Delta E \times m) \quad (1)$$

where q, m, C, and ΔE are the corresponding stored charge, mass of PC NPs, specific capacitance, and potential range during the CV operation, respectively. Based on the above equation, the mass ratio of each electrode (positive and negative electrodes composed of 20.5 bilayers) was adjusted to $m$MnO/$m$Fe₃O₄≈0.79. Accordingly, the area ratio of each electrode was designed to be 1.58:2 (MnO/Fe₃O₄).

*Aqueous ASC*: First, the formed electrodes were fixed on glass slides, then sufficiently soaked in a 1 M Na₂SO₄ electrolyte. Sequentially, both electrodes were assembled with a 25 μm thick separator (Celgard 3501) sandwiched between two electrodes.

*All-solid-state ASC*: For the solid-state ACSs, PVA/Na₂SO₄ polymer gel was employed as both the separator and electrolyte. In a typical preparation, 1 M Na₂SO₄ and 6 g of PVA (Sigma-Aldrich) were mixed with 60 ml of deionized water at 90 °C for 1 h. After the Na₂SO₄/PVA gel became clear, the as-prepared positive and negative electrodes were immersed for 1 h, then dried under vacuum for 30 min. After that, the electrodes were assembled facing each other.

*Electrochemical measurement*: The electrochemical characteristics of all MP-SC electrodes were investigated using an Ivium-n-Stat (Ivium Technologies). In the three-electrode configuration used to characterize the individual electrodes, a Pt wire, a Ag/AgCl (saturated by 3 M NaCl), and the MP-SC electrode (active area of 2 cm²) were used as the counter, reference, and working electrode in 1 M Na₂SO₄ electrolyte, respectively. CV and GCD measurements were performed with potentials ranging from 0 to ±1 V for the three-electrode system and from 0 to 2 V for the two-electrode cell. EIS was conducted for the MP-based electrodes over a frequency range of 100 kHz to 0.1 Hz with a perturbation amplitude of 0.01 V. The obtained impedance spectra were processed by ZView software (Scribner Associates Inc.). Error bars in all electrochemical data (areal or mass capacitance and EIS) represent the s.d. of the each values collected from three samples.

*Electrochemical performance evaluation*: The electrochemical capacitance of the formed MP-based SC electrodes was evaluated by the following equations[66]:

$$\text{Specific capacitance } (C) = \frac{\int i(V)dV}{2v\Delta VS} \text{(for CV)} \quad (2)$$

$$\text{Specific capacitance } (C) = \frac{I\Delta t}{\Delta V}. \text{ (for galvanostatic charge/discharge measurement)} \quad (3)$$

where i, v, ΔV, I, and Δt represent the current, scan rate (mV s⁻¹), operating voltage window, applied current density, and discharge time, respectively. The variable S in Eq. (2) indicates the mass of the active materials or active electrode area. The power and energy density of the MP-ASCs were determined from galvanostatic discharge profiles using the following equations:

$$\text{Energy density } (E) = \frac{1}{2}CV^2 \quad (4)$$

$$\text{Power density } (P) = \frac{V^2}{4RS} \quad (5)$$

Where C and R are the total capacitance and the resistance of the MP-ASC, respectively. The variable R was obtained from ΔiR/2i. Here, ΔiR and i represent the voltage drop between the first two points in the discharge profile and the applied current, respectively.

**Data availability**. The authors declare that the data supporting the findings of this study are available from the corresponding author on request.

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

## Acknowledgements

This work was supported by the National Research Foundation (NRF) grant funded by the Korean government (NRF-2015R1A2A1A01004354 and NRF-2016M3A7B4910619).

## Author contributions

Y.K., S.W.L. and J.C. conceived and designed the experiments. Y.K., M.K., W.K.B. and B.L. conducted experiments. Y.K., S.W.L. and J.C. co-wrote the manuscript.

## Additional information

**Competing interests:** The authors declare no competing financial interests.

