## [Peer Review File · Nature Communications]

Reviewers' comments:

Reviewer #1 (Remarks to the Author):

The subject is timely and of great interest. The authors designed and performed a nice set of novel experiments to realize high performance supercapacitors with metal-like cellulose papers. The results are interesting and useful. The discussion part is of in depth. The paper was well written. The reviewer believes that this is a paper of high quality. It is suggested that the paper be accepted after the following comments are taken care of.

1. Authors need to compare the results with other flexible, metal-oxide graphene hybrid supercapacitors. Cotton textile derived activated carbon fibers have been used to construct flexible supercapacitors. Comparison is essential. The authors need to discuss this with reference to the following papers.

Microstructural design of hybrid CoO@NiO and graphene nano-architectures for flexible high performance supercapacitors, *Journal of Materials Chemistry A*, 3 (2015) 14833-14844

Cotton textile enabled, all-solid-state flexible supercapacitors, *RSC Advances*, 5 (2015) 15438-15447

2. Chemically synthetic, conductive carbon fibers have been used as flexible backbones to construct supercapacitors. The authors need to discuss the advantages of using metal-like cellulose papers over other fibers/textiles. The following papers should help such discussion.

Flexible Zn₂SnO₄/MnO₂ Core/Shell Nanocable– Carbon Microfiber Hybrid Composites for High-Performance Supercapacitor Electrodes, *Nano letters* 11 (2011) 1215-1220

Flexible All-Solid-State Hierarchical NiCo₂O₄/Porous Graphene Paper Asymmetric Supercapacitors with an Exceptional Combination of Electrochemical Properties, *Nano Energy* 13 (2015) 306–317

3. Biomass-derived renewable carbon materials with porous architecture have shown eminent electrochemical properties. Are metal-like cellulose papers better than biomass activated carbon? The authors need to extend the discussion with reference to the following papers.

Biomass-derived renewable carbon materials for electrochemical energy storage, *Materials Research Letters* 5 (2017) 69–88

High-performance supercapacitors and batteries derived from activated banana-peel with porous structures, *Electrochimica Acta* 222 (2016) 1257–1266

Reviewer #2 (Remarks to the Author):

The manuscript presents a supercapacitor design based on mediated layer-by-layer assembly of metal nanoparticles on a variety of surfaces. The manuscript calls particular attention to the energy density, power density, and flexibility of the realized cells.

The highlight of the manuscript is the assembly process, which is useful for not only super capacitors but batteries and any other required “printed” conductors. The manuscript should emphasize the structural and conductivity fundamentals of these electrodes rather than the device level

performance.

Much attention in the manuscript is focused on the performance of the capacitors. While important, the comparisons to other work and in particular commercial cells are a bit misplaced. From what I can tell the results presented herein are based on active material basis, and comparisons are made to system level commercial cells. Aside from this, the true impact of this work will flow from the process, and the attention provided to the performance of the cell will have less impact over time.

A bit more discussion would be helpful regarding challenges with mass loading. Electrochemical and mechanical cycling results are presented for relatively low mass loadings (e.g. 0.95 to 1.5 mg/cm²) when results for 4 mg/cm² are also presented.

REVIEWERS' COMMENTS:

Reviewer #1 (Remarks to the Author):

The revised version is of much better quality. The authors fully addressed my concerns and made significant improvements. I suggest that the revised manuscript be accepted as is.

Reviewer #2 (Remarks to the Author):

All questions addressed

Seung Woo Lee
Assistant Professor
George W. Woodruff School of Mechanical Engineering
Georgia Institute of Technology, Atlanta, GA 30332

Telephone: +1-404-385-0764
E-mail: seung.lee@me.gatech.edu

May 4th, 2017

Dr. Yaoqing Zhang
Editor: *Nature Communications*

Dear Dr. Yaoqing Zhang,

Thank you for forwarding the reviewers' comments on our manuscript submission to *Nature Communications*, "Flexible Supercapacitor Electrodes Based on Real Metal-like Cellulose Papers". The positive comments of all reviewers are greatly appreciated, and we have found the reviewers' critiques to be very helpful for the improvement of the paper. We are pleased to submit a revised version of the paper that incorporates changes in response to the reviewers' comments. Please find specific responses to each of the comments as detailed below.

Sincerely,

Seung Woo Lee

Jinhan Cho

Response to Reviewers' Comments:

Referee: 1

Comments to the Author: *The subject is timely and of great interest. The authors designed and performed a nice set of novel experiments to realize high performance supercapacitors with metal-like cellulose papers. The results are interesting and useful. The discussion part is of in depth. The paper was well written. The reviewer believes that this is a paper of high quality. It is suggested that the paper be accepted after the following comments are taken care of.*

Response: We appreciate the reviewer's positive comments on the importance of our work. We here address the reviewer's comments as detailed below.

Comment #1. *Authors need to compare the results with other flexible, metal-oxide graphene hybrid supercapacitors. Cotton textile derived activated carbon fibers have been used to construct flexible supercapacitors. Comparison is essential. The authors need to discuss this regarding the following papers. 'Microstructural design of hybrid CoO@NiO and graphene nano-architectures for flexible high performance supercapacitors, Journal of Materials Chemistry A, 3 (2015) 14833-14844' 'Cotton textile enabled, all-solid-state flexible supercapacitors, RSC Advances, 5 (2015) 15438-15447'*

Response: We appreciate the reviewer's suggestion. We incorporated these references in the introduction part and also compared the performances in the result section.

Page 3

"Recently, aside from these conductive substrates, a variety of pseudocapacitive (PC) materials with higher capacitances have been introduced to improve the energy density.^{14-17,}"

Page 17

"In this case, the MP-ASCs exhibited maximum power and energy densities of 129 kW·kg⁻¹ and 121 W·h·kg⁻¹ for the liquid-state MP-ASC and 95.1 kW·kg⁻¹ and 81.7 W·h·kg⁻¹ for the solid-state MP-ASC, respectively, even at high mass loadings of the active energy materials, which are superior to the other previously reported flexible SCs (Supplementary Table S2)^{14-17,S12-S22}."

References

14. Gao, Z., Song, N. & Li, X. Microstructural design of hybrid CoO@NiO and graphene nano-architectures for flexible high performance supercapacitors. *J. Mater. Chem. A*, **3**, 14833-14844 (2015).
15. Gao, Z., Song, N., Zhang, Y. & Li, X. Cotton textile enabled, all-solid-state flexible supercapacitors. *RSC Adv.* **5**, 15438-15447 (2015).
16. Bao, L., Zang, J. & Li, X. Flexible Zn₂SnO₄/MnO₂ core/shell nanocable-carbon microfiber hybrid composites for high-performance supercapacitor electrodes. *Nano lett.* **11**, 1215-1220 (2011).
17. Gao, Z. *et al.* Flexible all-solid-state hierarchical NiCo₂O₄/porous graphene paper asymmetric supercapacitors with an exceptional combination of electrochemical properties. *Nano Energy* **13**, 306-317 (2015).

Comment #2. *Chemically synthetic, conductive carbon fibers have been used as flexible backbones to construct supercapacitors. The authors need to discuss the advantages of using metal-like cellulose papers over other fibers/textiles. The following papers should help such discussion. 'Flexible Zn₂SnO₄/MnO₂ Core/Shell Nanocable- Carbon Microfiber Hybrid Composites for High-Performance Supercapacitor Electrodes, Nano letters 11 (2011) 1215-1220' 'Flexible All-Solid-State Hierarchical NiCo₂O₄/Porous Graphene Paper Asymmetric Supercapacitors with an Exceptional Combination of Electrochemical Properties, Nano Energy 13 (2015) 306-317'*

Response: We appreciate the reviewer's suggestion. First of all, we incorporated the suggested references in the introduction and result parts (please see the response for the comment #1). In addition, we have added a discussion on the advantage of the cellulose paper compared to the carbon fiber substrates on page 9.

Page 9

“It should be noted that the electric conductivity of metal NP layers is basically superior to those of carbon-based materials. Additionally, the packing density of PC NPs as well as metal NPs shown in our approach is much higher than those of NPs formed by other approaches. Therefore, metallic paper-based SC devices can significantly improve the power and energy densities compared to the previously reported carbon-based SC devices.^{16,17”}

References

16. Bao, L., Zang, J. & Li, X. Flexible Zn₂SnO₄/MnO₂ core/shell nanocable-carbon microfiber hybrid composites for high-performance supercapacitor electrodes. *Nano lett.* **11**, 1215-1220 (2011).
17. Gao, Z. *et al.* Flexible all-solid-state hierarchical NiCo₂O₄/porous graphene paper asymmetric supercapacitors with an exceptional combination of electrochemical properties. *Nano Energy* **13**, 306–317 (2015).

Comment #3. *Biomass-derived renewable carbon materials with porous architecture have shown eminent electrochemical properties. Are metal-like cellulose papers better than biomass activated carbon? The authors need to extend the discussion regarding the following papers. ‘Biomass-derived renewable carbon materials for electrochemical energy storage, Materials Research Letters 5 (2017) 69–88’ ‘High-performance supercapacitors and batteries derived from activated banana-peel with porous structures, Electrochimica Acta 222 (2016) 1257–1266’*

Response: We appreciate the reviewer's suggestion on improving the manuscript. Biomass-derived renewable carbon materials are attractive candidates for energy storage applications especially for large-scale applications. Compared to the biomass-derived carbons, the metal-like cellulose papers have unique advantages in developing microelectronics or microscale energy storage devices, including pin-point control over the thickness or energy density as well as excellent electric conductivity. Therefore, if our approach allowing densely packed metal and PC NP layers is applied to biomass-derived carbon materials with highly porous 3D structure, we believe that power and energy density of supercapacitor devices can be further enhanced. In this view, we have added the following discussion to the conclusion section on page 18 in the revised manuscript.

Page 19

“We also envision that this approach can be extended to the biomass-derived carbon materials^{59,60} with a highly porous architecture for developing large-scale energy storage devices with high performance.”

References

59. Gao, Z., Zhang, Y., Song, N. & Li, X. Biomass-derived renewable carbon materials for electrochemical energy storage. *Mater. Res. Lett.* **5**, 69–88 (2017).
60. Zhang, Y., Gao, Z., Song, N. & Li, X. High-performance supercapacitors and batteries derived from activated banana-peel with porous structures. *Electrochim. Acta* **222**, 1257–1266 (2016).

Referee: 2

Comments to the Author: *The manuscript presents a supercapacitor design based on mediated layer-by-layer assembly of metal nanoparticles on a variety of surfaces. The manuscript calls particularly attention to the energy density, power density, and flexibility of the realized cells.*

The highlight of the manuscript is the assembly process, which is useful for not only super capacitors but batteries and any other required “printed” conductors. The manuscript should emphasize the structural and conductivity fundamentals of these electrodes rather than the device level performance.

Much attention in the manuscript is focused on the performance of the capacitors. While important, the comparisons to other work and in particularly commercial cells are a bit misplaced. From what I can tell the results presented herein on based on active material basis, and comparisons are made to system level commercial cells. Aside from this, the true impact of this work will flow from the process, and the attention provided to the performance of the cell will have less impact over time.

A bit more discussion would be helpful regarding challenges with mass loading. Electrochemical and mechanical cycling results are presented for relatively low mass loadings (e.g. 0.95 to 1.5 mg/cm²) when results for 4 mg/cm² are also presented.

Response: We appreciate the reviewer’s positive comments on the importance of our work. We here address the reviewer’s comments as detailed below.

Comment #1. *The highlight of the manuscript is the assembly process, which is useful for not only super capacitors but batteries and any other required “printed” conductors. The manuscript should emphasize the structural and conductivity fundamentals of these electrodes rather than the device level performance.*

Response: We appreciate the reviewer’s valuable comment. In this study, the metallic paper (MP) electrodes were realized by introducing the (TREN/TOABr-Au NP)_n multilayers onto the insulating cellulose paper with a highly porous structure using a molecular-mediated LbL assembly in organic media. In this case, the formed Au NP multilayers were densely but uniformly coated to the interior and exterior of the cellulose paper without any NP agglomeration, which allowed the high electrical conductivity while preserving the porous nature of the cellulose paper (Fig. 2d and S8). Additionally, the LbL-assembled TREN/TOABr-Au NP multilayers exhibited the dense and disordered internal structure with a large amount of nanopores formed among Au NPs (**Fig. R1**) as well as the macroporous structure of the cellulose paper. That is, the formed TREN/TOABr-Au NP multilayer films displayed a mass density of 13.2 g·cm⁻³ (the mass density of bulk gold ~ 19.1 g·cm⁻³), a porosity of 30.8 %, and a pore volume of 0.022 cm³·g⁻¹, respectively (**Fig. R2**). This structural uniqueness of MP electrodes facilitates the ion transfer during the electrochemical operation, enabling high rate capability of the MP-SC electrodes.

Furthermore, the (TREN/TOABr-Au NP)_n multilayers exhibit the high electrical conductivity due to the significantly short inter-distance between the adjacent Au NPs. It should be noted that only one molecule linker ‘TREN’ exists between the neighboring NPs, facilitating electron communication. To investigate the electrical conduction mechanism of the multilayers, the electrical conductivity was monitored under temperature variation from 2 to 300 K (Fig. S5). In this case, the temperature dependence of the conductivity (σ) did not correspond to the linear dependence with the variable-range charge transport equation ($\sigma = \sigma_0 \exp(-A/T^{1/d+1})$) for hopping ($d = 3$) or tunneling ($d = 1$) conduction processes. On the other hand, the temperature coefficient of the electrical resistivity was measured to be a positive value of $1.64 \times 10^{-3} \text{ K}^{-1}$, which is the characteristic of pure metals, suggesting the typical metallic behavior.

For more clarifying these issues raised by reviewer 2, Figure R1, R2, and discussion part have been added to Supplementary Information (**Figure S3** and **S4**) and main text in our revised manuscript, respectively.

On Figure S3 in Supplementary Information:

Figure S3. **b**, Tilted SEM and **c**, AFM topographic images of (TREN/TOABr-Au NP)_{30.5} multilayers deposited onto the silicon wafer substrate.

On Figure S4 in Supplementary Information:

Figure S4. A plot of areal mass ($\text{mg}\cdot\text{cm}^{-2}$) vs. film thickness (nm) of (TREN/TOABr-Au NP)_n multilayers deposited on QCM electrodes (left) and thickness profiles at two different points measured using AFM (right). The slope in the graph indicates the mass density of multilayers ($13.2\text{ g}\cdot\text{cm}^{-3}$). Additionally, the porosity of the multilayer film was 30.9 %, which was calculated by the following equation: porosity (%) = $(1 - (\rho_{\text{film}}/\rho_{\text{bulk}}) \times 100$, where ρ is the mass density.

Page 6

“Furthermore, these Au NP multilayers exhibited the dense and disordered structure with a large amount of nanopores formed among Au NPs as well as macroporous structure of the cellulose paper. That is, the (TREN/TOABr-Au NP)_{30.5} multilayers had a high mass density of 13.2 g·cm⁻³ (the mass density of bulk gold ~ 19.1 g·cm⁻³), a porosity of 30.8 %, and a pore volume of 0.022 cm³·g⁻¹, respectively. As a result, it was considered that this structural uniqueness of MP electrodes could facilitate the ion transfer, which is closely related to the power performance of MP-SC.”

Page 7

“To investigate the electron transfer mechanism of the (TREN/TOABr-Au NP)_n multilayers, the temperature dependent electrical conductivity was monitored by four probe measurement over a temperature range of 2 – 300 K (Supplementary Fig. S5). In this case, the temperature dependence of conductivity did not correspond to the linear dependence with the following equation for semiconducting kinetics: $\sigma = \sigma_0 \exp(-A/T^{1/d+1})$ for variable-range hopping (VRH) ($d = 3$) or tunneling ($d = 1$) mechanism, where σ is the conductivity, T is the absolute temperature (K), A is a constant, and d is the dimensionality^{37,48}.”

Page 8

“In this case, the formed Au NP multilayers show a positive temperature coefficient of $1.64 \times 10^{-3} \text{ K}^{-1}$ obtained by following relationship: $\Delta R/R_0 = \alpha \Delta T$, where R and α are the resistance (Ω) and the temperature coefficient, respectively.”

Comment #2-1. Much attention in the manuscript is focused on the performance of the capacitors. While important, the comparisons to other work and in particularly commercial cells are a bit misplaced. From what I can tell the results presented herein on based on active material basis, and comparisons are made to system level commercial cells.

Response: We appreciated the reviewer’s comment. We agree with the reviewer that the comparison of our results with the system-level commercial cells is not appropriate as commercial supercapacitors or batteries exhibit their performances based on a whole device mass. Therefore, we removed the comparison with conventional devices in the Ragone plot. The revised Ragone plot now only compares the performance of our supercapacitors with other reported works based on the mass of the active materials as shown in Fig. 5e.

On Figure 5e:

Figure 5e. Ragone plots of the solid- and liquid-state MP-ASCs as a function of the total active energy materials.

Comment #2-2. *Aside from this, the true impact of this work will flow from the process, and the attention provided to the performance of the cell will have less impact over time.*

Response: We appreciated the reviewer's helpful comment. According to reviewer's comment, we have added the following discussion on the advantages of our process to our manuscript:

Page 9

“It should be noted that the electric conductivity of metal NP layers is basically superior to those of carbon-based materials. Additionally, the packing density of PC NPs as well as metal NPs shown in our approach is much higher than those of NPs formed by other approaches. Therefore, metallic paper-based SC devices can significantly improve the power and energy densities compared to the previously reported carbon-based SC devices.^{16,17”}

Page 11

“In should be noted that the excellent charge storage behavior of the (TREN/OA-MnO NP)_n-coated MP electrodes with high mass loading can be achieved not only by structural benefits of highly porous cellulose paper, but also by effective removal of insulating bulky ligands (i.e., TOABr or OA) bound to the surface of Au NPs during molecule-mediated LbL assembly process. That is, the ligand exchange from electrochemically nonactive and bulky ligands to hydrophilic TREN molecules allows the low charge transfer resistance, better penetration of the electrolyte, and enhanced utilization of active materials.”

References

16. Bao, L., Zang, J. & Li, X. Flexible Zn₂SnO₄/MnO₂ core/shell nanocable-carbon microfiber hybrid composites for high-performance supercapacitor electrodes. *Nano lett.* **11**, 1215-1220 (2011).

17. Gao, Z. *et al.* Flexible all-solid-state hierarchical NiCo₂O₄/porous graphene paper asymmetric supercapacitors with an exceptional combination of electrochemical properties. *Nano Energy* **13**, 306–317 (2015).

Comment #2-1. *A bit more discussion would be helpful regarding challenges with mass loading. Electrochemical and mechanical cycling results are presented for relatively low mass loadings (e.g. 0.95 to 1.5 mg/cm²) when results for 4 mg/cm² are also presented.*

Response: We appreciate the reviewer's comment. The high mass loading of active materials enables the high areal capacity. Therefore, this is a critical design factor in the textile-based flexible/wearable energy storage systems for the practical applications such a smart garments. However, typically, as the loading amount of transition metal oxide in electrodes increases, the electrochemical response tends to become insensitive due to the semiconducting nature of metal oxide. To overcome this problem, we further inserted the metal layers into the metal oxide NP-based multilayers, resulting in the improved electrochemical responses by reducing the internal resistance of the electrodes.

According to reviewer's comment, we have included additional electrochemical analysis for high mass loading of 4.09 mg·cm⁻², which showed negligible capacity decrease up to 5,000 cycles in Figures 4f and 5f. We added the more discussion relating to mass dependent electrochemical response to the revised manuscript and supplementary information.

Page 13

“The (TREN/OA-MnO NP)_n-coated MP electrodes with 0.23, 1.56, and 4.09 mg·cm⁻² maintained 88.9, 91.8, and 89.6 % of their initial capacitances, respectively, after 5,000 cycles during CV at 100 mV·s⁻¹ (Fig. 3f).”

Page 14

“Additionally, the alternating MnO/Au NP electrodes exhibited improved capacitive behaviors compared to the (TREN/OA-MnO NP)_n electrodes at the scan rate of 5 – 500 mV·s⁻¹ (Supplementary Fig. S21).”

Page 15

“More effective charge transport phenomena of the alternating MnO/Au NP electrodes can be further clarified by the loading dependent electrochemical behavior with different scan rate (Supplementary Fig. S22). In the case of the alternating MnO/Au NP electrodes, by increasing the mass loading, the areal capacitance gradually increased until the fast scan rate of $500 \text{ mV}\cdot\text{s}^{-1}$ even at the mass loading of $1.56 \text{ mg}\cdot\text{cm}^{-2}$ (electrode area is $1 \text{ cm} \times 2 \text{ cm}$). Although the slight performance fading is observed for high mass loading of $4.09 \text{ mg}\cdot\text{cm}^{-2}$ from a scan rate of $200 \text{ mV}\cdot\text{s}^{-1}$, these rate performance is superior to that of the (TREN/OA-MnO NP)_n-coated MP electrode without additional Au NP layers, indicating much better charge transport behavior.”

Page 15

“~ the alternating MnO/Au NP electrodes with mass densities of 0.95, 1.56, and $4.09 \text{ mg}\cdot\text{cm}^{-2}$ maintained 93, 96.4, and **91.3** % of their initial capacitance after 5,000 cycles, respectively (Fig. 4f).”

Figure 3:

Figure 3e and 3f. e, Scan rate dependence of the specific mass capacitance of the $(TREN/OA-MnO NP)_n$ -coated MP electrodes with difference mass densities. f, Cycling retention of the $(TREN/OA-MnO NP)_n$ -coated MP electrodes with mass densities of 0.23, 1.56, and 4.09 $mg \cdot cm^{-2}$, respectively.

On Figure 4:

Figure 4e and 4f. e, Specific mass capacitance of the alternating MnO/Au NP (with Au layer) and the $(TREN/OA-MnO NP)_n$ (without Au layer)-coated MP electrodes with different mass densities as a function of the scan rate, respectively. f, Cycling retention of alternating MnO/Au NP electrodes with mass densities of 0.95, 1.56, and 4.09 $mg \cdot cm^{-2}$, respectively.

On Figure S14 in Supplementary Information:

Figure S14. a, Nyquist plots of the (TREN/OA-MnO NP)_n multilayer-coated MP-SC electrodes as a function of the mass loading of MnO NPs.

On Figure S15 in Supplementary Information:

Figure S15. CV response of MN-based (TREN/OA-MnO NP)_n electrodes with different mass densities of MnO NPs: **a**, 0.23, **b**, 0.45, **c**, 0.95, **d**, 1.56, and **e**, 4.09 mg·cm⁻² as the scan rate increased from 5 to 500 mV·s⁻¹, respectively.

On Figure S20 in Supplementary Information:

Figure S20. Nyquist plot of the alternating MnO/Au NP electrodes with increasing the mass density **from 0.23 to 4.09 $\text{mg}\cdot\text{cm}^{-2}$** , respectively. The inset indicates the high frequency region of spectra.

On Figure S21 in Supplementary Information:

Figure S21. CV scans of the alternating MnO/Au NP electrodes with different mass densities of **a**, 0.23, **b**, 0.45, **c**, 0.95, **d**, 1.56, and **e**, 4.09 $\text{mg}\cdot\text{cm}^{-2}$ at a scan rate of 5 – 500 $\text{mV}\cdot\text{s}^{-1}$, respectively. In this case, the CV responses of the alternating MnO/Au NP electrodes show the larger and less distorted than those of (TREN/OA-MnO NP)_n electrodes shown in Fig. S15, demonstrating the better charge transport behavior.

On Figure S22 in Supplementary Information:

Figure S22. Mass dependent areal capacitance of **a**, the alternating MnO/Au NP electrodes and **b**, (TREN/OA-MnO NP)_n electrodes with various scan rate of 5 to 500 mV·s⁻¹.

Response to Reviewers' Comments:

Reviewer #1 (Remarks to the Author):

The revised version is of much better quality. The authors fully addressed my concerns and made significant improvements. I suggest that the revised manuscript be accepted as is.

Response: We appreciate the reviewer's comments to improve the novelty and importance of our work.

Reviewer #2 (Remarks to the Author):

All questions addressed

Response: We greatly appreciate the reviewer's encouragement.